# Biliary Diseases from the Microbiome Perspective: How Microorganisms Could Change the Approach to Benign and Malignant Diseases

**DOI:** 10.3390/microorganisms10020312

**Published:** 2022-01-28

**Authors:** Cecilia Binda, Giulia Gibiino, Chiara Coluccio, Monica Sbrancia, Elton Dajti, Emanuele Sinagra, Gabriele Capurso, Vittorio Sambri, Alessandro Cucchetti, Giorgio Ercolani, Carlo Fabbri

**Affiliations:** 1Gastroenterology and Digestive Endoscopy Unit, Forlì-Cesena Hospitals, Ausl Romagna, 47121 Forlì, Italy; cecilia.binda@auslromagna.it (C.B.); giulia.gibiino@auslromagna.it (G.G.); monica.sbrancia@auslromagna.it (M.S.); elton.dajti2@unibo.it (E.D.); carlo.fabbri@auslromagna.it (C.F.); 2Department of Medical and Surgical Sciences–DIMEC, Alma Mater Studiorum–University of Bologna, 90015 Bologna, Italy; alessandro.cucchett2@unibo.it (A.C.); giorgio.ercolani@auslromagna.it (G.E.); 3Endoscopy Unit, Fondazione Istituto San Raffaele-G. Giglio, 90015 Cefalù, Italy; emanuelesinagra83@googlemail.com; 4Euro-Mediterranean Institute of Science and Technology (IEMEST), 90100 Palermo, Italy; 5Division of Pancreato-Biliary Endoscopy and EUS, Pancreas Translational and Clinical Research Center, San Raffaele Scientific Institute IRCCS, 20132 Milano, Italy; capurso.gabriele@hsr.it; 6Unit of Microbiology, The Great Romagna Hub Laboratory, 47522 Pievesestina, Italy; vittorio.sambri@auslromagna.it; 7Unit of Microbiology, Department of Pathological Anatomy, Trasfusion Medicine and Laboratory Medicine, University of Bologna, 40125 Bologna, Italy; 8Department of General and Oncologic Surgery, Morgagni-Pierantoni Hospital, Ausl Romagna, 47121 Forlì, Italy

**Keywords:** biliary tract microbiome, cholangiocarcinoma, biliary cancer, cholangitis, oncobiome

## Abstract

Recent evidence regarding microbiota is modifying the cornerstones on pathogenesis and the approaches to several gastrointestinal diseases, including biliary diseases. The burden of biliary diseases, indeed, is progressively increasing, considering that gallstone disease affects up to 20% of the European population. At the same time, neoplasms of the biliary system have an increasing incidence and poor prognosis. Framing the specific state of biliary eubiosis or dysbiosis is made difficult by the use of heterogeneous techniques and the sometimes unwarranted invasive sampling in healthy subjects. The influence of the microbial balance on the health status of the biliary tract could also account for some of the complications surrounding the post-liver-transplant phase. The aim of this extensive narrative review is to summarize the current evidence on this topic, to highlight gaps in the available evidence in order to guide further clinical research in these settings, and, eventually, to provide new tools to treat biliary lithiasis, biliopancreatic cancers, and even cholestatic disease.

## 1. Introduction: The Microbiome in Healthy Patients

The human microbiota is one of the densest, and quickly developing ecosystems [1]. The term “microbiota” usually defines the assemblage of living microorganisms present in certain environment [2]. However, as phages, viruses, plasmids, prions, viroids, and free DNA are usually not considered as living microorganisms [3], they do not belong to the microbiota. The term microbiome, as it was originally postulated by Whipps and colleagues [4], includes not only the community of the microorganisms but also their “theatre of activity” [5].

In healthy conditions, the microbiome bacteria interact with the epithelial barrier and with the immune system influencing their feedback, but they are also able to produce substances that influence the local metabolism, thus maintaining homeostasis [6,7]. For example, the normal gut microbiome has a specific function in host nutrient metabolism, xenobiotic and drug metabolism, maintenance of the structural integrity of the bowel mucosal barrier, immunomodulation, and defense from pathogens [8]. Therefore, an imbalance of the gut microbiome, due to antibiotics or to bacterial translocation, can promote the development of diseases [7].

The gastrointestinal system is one of the largest storing places of microorganisms in the human body and it holds both commensal and pathogenic microbial species [5,9]. To date, little is known about the composition of the biliary tract microbiota and its influence on the development of biliary diseases. In fact, bile represents a biological fluid produced in the liver, stored in the gallbladder (interdigestive), and successively released into the duodenum after food ingestion [10]. While the microbial species of different parts of gastrointestinal system have been extensively investigated in health and disease, the identification of bile microbiota has not been addressed [10]. Furthermore, the few available data on the biliary microbiota are limited to experimental animal and human pathological models [11].

In their experimental study, Jiménez and co-workers analyzed the bile, gallbladder mucus, and mucosal microbiome of healthy pigs using both culture-based and metagenomics techniques [10]. All the cultured samples harvested bacterial species and the number of identified species ranged from 3 to 20 per sample. All the bacteria isolated from cultures were broadly balanced among the Firmicutes (34%), Actinobacteria (32%) and Proteobacteria (32%) phyla. Bacteroidetes accounted for only a smaller part (2% of the isolates), thus highlighting an inadequate adaptation to the biliary environment. On the other hand, at the genus level, *Staphylococcus*, *Streptococcus*, *Kocuria*, *Rothia*, *Acinetobacter*, and *Psychrobacter* were isolated from different samples, suggesting their possible role as members of the core biliary microbiota of pigs [10]. Interestingly, the microbiological analysis of gallbladder mucus and mucosa broadened the spectrum of bacteria present in the bile that could possibly colonize these niches [10].

As aforementioned, research on the composition of the “healthy” human biliary microbiota is challenging. Indeed, bile sampling techniques, such as endoscopic retrograde cholangiopancreatography (ERCP), percutaneous biliary drainage, and surgical sampling, are invasive procedures that are typically performed when a biliary tract disease is diagnosed or suspected.

In the study performed by Molinero et al., the biliary microbiota of 27 liver donors (13 without and 14 with cholelithiasis) were analyzed [12]. The 16S ribosomal RNA sequencing showed a prevalence of Actinobacteria, Firmicutes and Bacteroidetes in both the bile samples and gallbladder tissues of subjects without gallstones [11,12]. Furthermore, a substantial increase in the presence of the Propionibacteriaceae family and *Sphingomonas* genus was also reported, when compared with individuals with gallstones [12]. This study provided evidence regarding the human biliary microbiota in healthy subjects. However, the study had limitations [2]. In fact, since liver donors undergo specific treatments and procedures, such as antibiotics, in the hospital’s intensive care units, their bile samples can hardly be considered “normal biliary microbiome”. In any case, these results give access to new perspectives in the identification of bacterial functions in a microbial ecosystem that was previously unexplored. Despite these interesting results, studies with a larger sample size are needed to confirm the findings. Indeed, the possible identification of a stable “biliary tract resident microbial community” may challenge the traditional knowledge on the development of biliary infectious diseases [11]. It may be tempting to hypothesize that a local dysbiotic process, rather than an ascending infection from the duodenum, better explains the occurrence of several biliary diseases, in a “microbiota-centric” view [11].

Recent insights into the biliary microbiota have improved the understanding of the pathogenesis of biliary diseases, such as gallstones, biliopancreatic cancer, and autoimmune cholangiopathies. The aim of this narrative review is to summarize the available evidence about the role of the biliary microbiome in the onset of gallstone disease, biliary cancer, autoimmune cholestatic diseases and in some settings, such as in liver transplantation (LT).

## 2. The Microbiome and Gallstone Disease

Biliary stones are a leading cause of hospitalization in gastroenterological departments worldwide; in Europe the prevalence of gallstone disease is increasing more and more, ranging between 5.9% and 21.9% in large surveys [13]. The European Association for the Study of the Liver (EASL) reports that in Europe about 20% of inhabitants suffers of lithiasis [14].

The possible role of bacterial microorganisms in gallstone formation has gained growing interest over the last decade. Its pathogenesis, with variations between countries and regions, implies the synergic effect of genetic and environmental factors, such as ethnicity, gender, age, lithogenic genes, hypersecretion of cholesterol or bilirubin by the gallbladder or liver, bile stasis, diet, metabolism, intestinal factors, the use of drugs, lifestyle, and comorbidities [15,16,17]. Gutiérrez-Díaz and coworkers [18] confirmed the association between diet, biliary microbiota and gallstone disease, demonstrating that in patients with lithiasis, a modification in the abundance of bile microorganisms occurs. Bacteroidaceae and *Bacteroides* appear to be negatively correlated with dairy product intake, and Bacteroidaceae, Chitinophagaceae, Propionibacteraceae, *Bacteroides*, and *Escherichia-Shigella* are positively correlated with several kinds of fiber, phenolics, and fatty acids. Changes of the dynamic equilibrium among all these factors may influence the risk of lithiasis occurrence.

Several studies have suggested that a resident biliary tract microbiome exists with a high level of similarity with the duodenal one. The intestinal bacteria (*Clostridium*, *Bifidobacterium*, *Peptostreptococcus*, *Bacteroides*, *Eubacterium*, and *Escherichia coli*) involved in bile acid metabolism can interfere with enterohepatic circulation, leading to lithogenesis [19,20], in particular Firmicutes, Proteobacteria, and Bacterioidetes [21]. Moreover, pathogenic bacteria of the oral cavity can play a promoter role for gallstone onset, influencing the motility of the gallbladder and the expression of mucin genes (MUC1, MUC3, and MUC4) [22,23].

In 1966, Maki et al. highlighted the connection between bacterial infection and the formation of pigmented gallstones for the first time [24], demonstrating that the inoculation of bacterial β-glucuronidase in the bile could hydrolyze the bilirubin glucuronide into bilirubin and glucuronic acid, which, precipitating with calcium, forms calcium bilirubinate. Indeed, β-glucuronidase-expressing bacteria have been frequently identified in the samples of patients with pigmented gallstones [25,26]. Other bacterial enzymes, such as phospholipases and bile acid hydrolases have later been shown to be implicated with similar mechanisms in the formation of pigmented gallstones [27,28], reported as containing bacterial sequences of *E. coli* and *Pseudomonas* sp. [29]. The following studies have confirmed Maki’s hypothesis, making this theory widely accepted. Even if the formation of cholesterol gallstones has traditionally been considered to be affected by metabolic imbalances and genetic variances rather than a bacterial damaging effect [15], over the years a growing literature emerged, demonstrating that alterations of the gut microbiota also contribute to the formation of cholesterol gallstones [22,30,31], especially after the advent of innovative genomic techniques. *Pseudomonas aeruginosa* and *Enterococcus faecalis* have been reported to shorten the cholesterol crystallization time in the bile, suggesting that these species may be crucial in the formation of cholesterol gallstones [32].

The gut microbiota plays an active role in bile acid metabolism, regulating the size and composition of bile acids [12,33]. Changes in the bile acid pool represent a leading etiopathogenetic factor, as bile is composed mainly of bile acids (nearly 50%); cholesterol and fatty acids account for nearly 20%, while phospholipids and bilirubin account for a minority of bile [34,35].

The biliary and gut microbiota are involved in almost all passages of bile formation, such as the metabolism of lipid and cholesterol, biotransformation, and enterohepatic circulation. Indeed, a perturbation of the gut microbiota may influence bile acid homeostasis in any step of the host metabolic pathways, in particular regarding glucose and cholesterol metabolism, which is crucial for gallstone genesis [36]. We report the main microorganisms involved in this setting in the gut and biliary microbiota in Table 1.

While the biliary system in healthy patients was once considered to be sterile [37], it has been demonstrated that the gallbladder physiologically has a composite microbiota, with different possible routes leading to bacterial colonization of the biliary system, including duodenal translocation trough the papilla or migration through blood vessels [38,39].

In 1995, Swidsinski et al. [40] analyzed the cholesterol gallstones from patients with negative bile culture using polymerase chain reaction (PCR)-based amplification and 16S ribosomal RNA sequencing, showing that bacterial DNA was present in 94% of mixed gallstones (containing cholesterol for 70–90%). Pure cholesterol gallstones (>90% of content) showed, instead, no bacterial DNA. Three groups of bacteria were identified: Propionibacteria-related, Clostridia-related, and Enterobacteria-related accounting for 45%, 35%, and 25% of the total, respectively.

While the high concentration of a single bacterial family is compatible with an infection, the concurrent presence of multiple bacterial species suggests a colonization. Several studies compared the biliary microbiota of patients with gallstones to those of other sites of the gastrointestinal tract with findings that all bacterial genera found in the biliary system were also retrieved in at least one other analyzed tract, thus supporting the case for bile colonization, which may be pivotal in lithiasis pathogenesis [41,42].

With the support of PCR, the presence of bacteria in the bile samples of patients affected by biliary tract disorders was discovered, with variable composition of *Capnocytophaga* spp., *Lactococcus* spp., *Bacillus* spp., *Staphylococcus haemolyticus*, *Enterobacter* or *Citrobacter* spp., *Morganella* spp., *Salmonella* spp., and *Helicobacter pylori* (*HP*) [42,43].

*HP*-produced urease represents a crucial link for gallstones through calcium precipitation [44]. One mechanism of action for *HP* is the release of proinflammatory and vasoactive substances, such as interleukins IL-1 and IL-6 and tumor necrosis factor (TNF)-alpha, which may cause oxidative stress and free radical release, that are related to gallbladder inflammatory disorders and to the cholelithogenesis [45,46].

The microbiota might also contribute indirectly to gallstone pathogenesis, affecting energy intake, bowel permeability, and supporting a chronic pro-inflammatory condition [47]. Several in depth studies investigated the metagenomic profiles of the different biliary bacterial communities. In patients with pigmented gallstones, genes extracted from *Klebsiella* and *Enterococcus* were found, which are thought to be involved in biofilm arrangement. Regarding cholesterol stones, bile resistance genes were selected from *Escherichia*, *Shigella*, *Serratia*, *Bacillus*, and *Klebsiella* [48].

Thanks to cultivation or PCR, several bacteria, such as *Escherichia coli*, *Klebsiella pneumoniae*, *Enterococcus faecium*, *Enterobacter cloacae*, and *Pseudomonas aeruginosa*, have been identified in bile or gallstone samples [49]. New methods, such as next-generation sequencing (NGS), helped us to significantly increase our knowledge regarding the microbial flora [50].

Wu and coworkers analyzed, in patients with cholesterol gallstone, gallbladder bile, gallstones, and feces by performing bacterial 16S rRNA amplicon sequencing [21]; they detected several gut bacterial operational taxonomic units (OTUs) in the biliary system and reported dysbiosis in the fecal samples from the patients with gallstones.

In a Chinese study, whole-metagenome shotgun (WMS) and 16S sequencing were used to investigate the bile samples of 15 patients with choledocholithiasis. Bile was collected from the common bile duct, and communities in the bile were compared to matched gut microbiota. Thirteen novel biliary bacterial species were identified in the human biliary tract that were never documented previously (such as *P. piscolens* and *Cellulosimicrobium cellulans*), revealing heterogeneity among individuals and a prevalence of microorganisms from the oral cavity/respiratory system instead of the bowel. To note, in this study there was no correlation between previous endoscopic sphincterotomy and the biliary microbial composition, in contrast with the hypothesis that sphincterotomy increases the risk of biliary tract infection [51].

In the previously cited study of Peng et al., they used culture-dependent and culture-independent methods to study, in 22 patients affected by cholesterol gallstone, the composition and function of bacterial clusters in cholesterol gallstones and bile [42]. It was reported that *Pseudomonas* spp. were the dominant inhabitants in both groups. Moreover, its major role in the formation of cholesterol stones is supported by showing that 30% of the culturable strains were able to secrete b-glucuronidase and phospholipase A2, and that the *Pseudomonas aeruginosa* strains had the highest enzyme activity.

Others have reported that the prevalence of *Pseudomonas*, *Bacillus*, *Klebsiella*, *Clostridium*, *Staphylococcus*, and *Enterobacter* is more relevant in cholesterol gallstones than in the bile, supporting the hypothesis that a dysfunction of the bowel barrier with subsequent bacterial translocation into the biliary tract may act as a critical point in lithogenesis [52]. Of the previously reported gena, *Escherichia*, *Brucella*, *Citrobacter*, *Shinella*, *Aurantimonas*, *Lachnospiraceae*, and *Lactobacillus* were founded in gallstones. Specifically, *Citrobacter*, *Lactobacillus*, and *Aurantimonas* are, indeed, common inhabitants of the gut. The presence of intestinal bacteria in cholesterol gallstones suggests that, after migration towards the gallbladder, the microorganisms might function as a trigger for the immune system and stimulate gallstone formation. Based on this study, four genera were potentially involved in cholesterol crystallization: *Pseudomonas*, *Enterococcus*, *Klebsiella*, and *Enterobacter*.

Furthermore, a recent study detected a higher prevalence of Bacteroidaceae, Prevotellaceae, Porphyromonadaceae, and Veillonellaceae in patients with gallstone disease [12] compared to individuals without diagnosed hepatobiliary pathology.

A different rate of cultured bacteria in the bile of patients with brown pigment lithiasis (between 53% and 100%), cholesterol gallstones (between 9% to 34%), and black pigment stones (from 9% to 19.6%) has been consistently reported [53,54,55]. A very recent study aimed to investigate the possible relation between the most abundant bile acids and the microbiota in gallstone disease [56]. A significant direct correlation between taurocholic acid (TCA) and taurochenodeoxycholicv acid (TCDCA) and the bile microbiota alpha-diversity was reported. In more detail, the presence of TCA was associated with species, such as *Jeotgalicoccus psychrophilus*, *Prevotella intermedia*, and *Haemophilus parainfluenzae* in the bile, while TCDCA concentration showed a positive link with the presence of *Microbacterium*, *Lutibacterium*, and *Sphingomonas* genera and *Prevotella intermedia* species. In conclusion, the different abundance of TCDCA and TCA correlates with bile microbiota alpha-diversity and the appearance of specific opportunistic pathogens in the bile of patients affected by gallstone disease.

Despite the availability of promising studies in this setting, it is still unknown if microbiological markers for gallstone disease exist, making it an interesting field for future research, especially for patients suffering from recurrent lithiasis. Besides its role as a biomarker of biliary diseases, the ‘gut microbiota–bile acid–host’ system may also offer a target to be manipulated, either with the use of specific probiotic strains or dietary interventions and prebiotics, as a novel strategy to handle the diseases associated with defective bile acid metabolism, in the attempt to restore a beneficial diversity of bacteria.

## 3. Microbiome and Biliary Cancer

The term “oncobiome” has been coined to describe the research field that studies how the microbiome is involved in the development of neoplastic disease [57]. Oncobiome research initially focused on colorectal cancer and in recent years has expanded into several other malignancies [58]. The possible mechanisms by which the microbiome influences cancer development include: (a) the role of bacterial toxins/metabolites on cancer onset and development [59,60,61], (b) how the microbiome can modulate the host’s local and systemic immune responses [62,63,64], and (c) specific changes in microbial and host metabolism [65,66,67]. It has been hypothesized that these host–microbe interactions can occur at both the local and systemic levels [68]. Herein, we provide an overview of the role of the microbiome in the development and progression of biliary cancer.

Cancers of the biliary tract encompass those arising from the intrahepatic and extrahepatic bile ducts, the gallbladder, and the ampulla of Vater and represent the sixth most common cause of malignant lesions in the gastrointestinal tract in western countries. We summarized the possible gut and biliary microbiota components involved in these diseases in Table 2.

### 3.1. Cholangiocarcinoma

Cholangiocarcinomas (CCA) are malignant neoplasms involving the intrahepatic or extrahepatic bile ducts and are usually classified, according to anatomy, into intrahepatic, perihilar, and distal cholangiocarcinomas. Given their high fatality rate and the silent progression of early disease, identifying the risk factors for the prevention and early detection of CCA is crucial. There are many known risk factors for the occurrence of CCA, including genetic background, chronic inflammation, environmental factors, and parasitic infestations, such as *Opisthorchis viverrini* (*OV*) and *Clonorchis sinensis*. However, the exact mechanisms of carcinogenesis remain unclear. Experimental studies have shown an association between microbial gut dysbiosis and the development of CCA [69,70,71], probably related to the bile acid metabolism pathway.

It has been reported that conjugated bile acids promote tumorigenesis, whereas unconjugated bile acids inhibit CCA cell proliferation. Therefore, an increase in either secondary or conjugated bile acids could be protumorigenic [70]. In a recent study, elevated plasma/stool ratios of conjugated secondary bile acids (glycoursodeoxycholic acid and tauroursodeoxycholic acid) were reported in CCA patients [72]. Moreover, in clinical studies conducted in the stools of CCA patients, a high richness (alpha-diversity) was reported compared to healthy individuals; in particular, a number of *Lactobacillus*, *Actinomyces*, Peptostreptococcaceae, *Alloscardovia*, and Bifidobacteriaceae were markedly increased. In addition, vascular invasion, which is a substantial prognostic factor, was associated with high levels of Ruminococcaceae species in the stools and higher circulating IL-4 levels [72]. Based on these results, it was speculated that the detection of plasma tauroursodeoxycholic acid, together with the presence of *Lactobacillus* and *Alloscardovia* in the stools, could be used as a potential diagnostic noninvasive biomarker for CCA [72]. Despite these data highlighting the existence of a certain association between the gut microbiota, the bile acid profile, and CCA, future studies with rigorous designs and investigations of patient outcomes are needed [73].

Conversely, little is known about the biliary microbiome of CCA patients and its possible role in tumorigenesis. Indeed, while changes in the bile microbiome in patients with CCA have been reported, whether this a cause or an effect of the tumor development is unknown. The biliary microbiome is conventionally investigated on bile samples collected during ERCP. Increased levels of biliary *Klebsiella pneumoniae* were positively correlated with CCA [74]. Moreover, the presence of *Helicobacter* spp., including non–*HP* species, is more frequent in the bile of CCA patients than in these with benign biliary disorders. Moreover, the most virulent cagA-1 HP strain is more abundant in the bile of CCA patients compared with that of cholelithitic or healthy individuals [75]. Based on these findings, it is possible to hypothesize a protumorigenic role for *Helicobacter* spp. through a pro-inflammatory cascade. Moreover, an assessment of the bile microbiota from ERCP samples by quantitative PCR showed an increase in species richness in the distal CCA patients compared with that of cholelithiatic patients. The most abundant species included *Gemmatimonadetes*, *Nitrospirae*, *Chloroflexi*, *Latescibacteria*, and *Planctomycetes* [76]. As for culture-independent methods, Avilés-Jiménez and coworkers [71] published the first report investigating microbiota in the bile ducts of patients with cholangiocarcinoma by NGS. In this study, Methylophilaceae, Sinobacteriaceae, *Actinomyces*, *Dialister*, *Novosphingobium*, *Prevotella*, *Fusobacterium*, and high-virulence *HP* were shown to be more abundant in the tumor tissue of CCA patients. A recent study investigated the microbiome of CCA and adjacent normal tissue and showed that Bifidobacteriaceae, Enterobacteriaceae, and Enterococcaceae were enriched in the tumor tissue specimens of CCA associated with liver fluke (*O. viverrini*) compared with CCA not associated with *O. viverrini*, suggesting a relationship between parasitic infections, the biliary microbiome, and the tissue microenvironment [77]. Saab et al. [78] recently analyzed the biliary microbiota of 28 CCA and 47 patients with gallstones, employed as control group. The most abundant genera in the CCA patients were *Enterococcus*, *Streptococcus*, *Bacteroides*, *Klebsiella*, and *Pyramidobacter*. The relative abundance of genera, such as *Bacteroides*, *Geobacillus*, *Meiothermus*, and *Anoxybacillus*, was also associated with specific comorbidities.

In conclusion, these results support the view that changes in the gut, bile microbiota, and in the biliary epithelium may play a significant role in CCA pathogenesis. Despite the variation in species abundance, *Helicobacter* spp., Bifidobacteriaceae, and Ruminococcaceae are generally increased in CCA patients. However, the field of microbiome investigation in CCA is new, and both basic and clinical investigations are needed to confirm whether the biliary microbiome may be an attractive biomarker or therapeutic target.

### 3.2. Gallbladder Cancer

Gallbladder cancer (GBC) is a highly invasive neoplasm with a 5-year survival rate of less than 5% [79]. Little is known about its etiopathogenesis. However, various factors, such as genetic susceptibility, infections, and lifestyle have been associated with the occurrence of GBC. The association between chronic calculous cholecystitis (CCC) and GBC has been reported [80,81]. Persistent biliary bacterial infections may expose individuals to the risk of an environment conducive to the development of malignant neoplasia. Similar to the intestinal mechanisms, the biliary mucosa contains several barriers, such as chemical, mechanical, and immunological barriers, that are responsible for immunological tolerance against commensals. Proteobacteria, Firmicutes, and Bacteroidetes mainly populate the gallbladder ecosystem [10]. Moreover, chronic colonization of *S. Typhi* may be a primary predisposing factor for the onset of GBC [82]. Song et al. [83] performed a metagenomic sequencing of the mucosal microbiome of CCC and GBC patients, exploring specific changes during the development of the malignant neoplasia form of chronic cholecystitis. In this study, the core microbiota was similar in both groups, whereas, during the development of GBC, species richness and evenness decreased, with differences also seen in the biliary microbial composition. In particular, *Peptostreptococcus stomatis*, *Fusobacterium mortiferum*, and *Enterococcus faecium* were found to be positively correlated with GBC. Future large studies with genomic analyses are needed to explore the interactions between the immune system and the host microbiome in the growth of GBC.

## 4. Microbiome and Autoimmune and Cholestatic Liver Diseases

### 4.1. Microbiota and Primary Biliary Cholangitis

Primary biliary cholangitis (PBC) is an uncommon chronic inflammatory autoimmune cholestatic liver disease that can have a serious course and progress to end-stage liver disease when untreated [84]. Its pathogenesis is multifaceted and occurs through the interaction between the immune and biliary pathways, where genetic and environmental factors play a substantial role [85]. As it often occurs in autoimmune disorders, it has been suggested that molecular mimicry between host antigens and microbes may act as a possible trigger [86]. More recently, the study of the gut–liver axis and, in particular, the farnesoid X receptor (FXR)–fibroblast growth factor (FGF)-19 signaling pathway and its role in bile acid metabolism, has led to a better understanding of PBC pathogenesis and the development of new targets for PBC treatment.

Despite these presumptions, only a few studies to date have evaluated the composition of gut microbiota in PBC patients [87,88,89,90], and they are reported in Table 3. Tang et al. [88] demonstrated in their pivotal paper that PBC patients presented a decreased gut microbiota richness; at the phylum level, *Bacteroides* spp. were significantly decreased and Fusobacteria and Proteobacteria spp were over-represented. The authors developed and validated a signature microbial signature based on 12 genera that could distinguish PBS patients from healthy controls with excellent accuracy; *Enterabacteriacea*, *Klebsiella*, *Pseudomonas*, *Clostridium*, *Veillonella*, *Streptococcus*, and *Haemophilus* were among the most relevant PBC-enriched genera. More recently, Chen and coworkers [91] elegantly captured the interplay between the bile acid profile, gut microbiota, and FGF-19 levels in PBC patients. Moreover, treatment with ursodeoxycholic acid (UDCA) reversed the abundance of most of the PBC-associated genera [88,89] and, more interestingly, the microbiota profile could predict the UDCA response, as non-responders presented a significantly lower abundance of the genus Faecalibacterium, a known butyrate-producing beneficial bacteria [92].

To our knowledge, only one study has evaluated the microbiome of the biliary tract in PBC patients [93]. Hiramatsu et al. aseptically extracted the gallbladder bile in patients with PBC undergoing liver transplantation and controls (patients with primary sclerosing cholangitis (PSC), hepatitis C virus, cholecystitis, etc.) and detected bacterial species by PCR amplification in 10/15 patients with PBC; the most common were Gram-positive cocci, such as *Staphylococcus aureus*, *Enterococcus faecium*, *Streptococcus pneumoniae* or other streptococci, and *Lactobacillus plantarum*, and they differed significantly from controls, including patients with cholecystolithiasis.

In conclusion, PBC patients present a distinct microbial profile, both in the gastrointestinal and in the biliary tract. Whether this difference is a consequence of the altered bile acid composition in PBC patients or plays a pathogenetic role in the development and progression of the disease is still to be determined. Future studies should investigate whether microbiota composition could predict the response to UDCA treatment, the severity of the disease, and the development of liver-related events in patients with PBC.

### 4.2. Microbiota and Autoimmune Hepatitis

Autoimmune hepatitis (AIH) is a chronic, progressive, and immunologically mediated inflammatory liver disorder [92]. Recent studies in animal models have proposed that increased intestinal permeability and gut microbiome dysbiosis play a role in the pathogenesis of liver inflammation in AIH [93,94]. Table 3 reports some of the main bacteria involved in this setting. Lin et al. [95] described, for the first time in humans, the features of a leaky gut (disruption of the architecture of the duodenal mucosa and reduced expression of tight junction proteins, such as ZO-1 and occludin) and dysbiosis, with a significant decrease of *Bifidobacterium* and *Lactobacillus* in AIH patients as compared to healthy controls. Since then, several authors have evaluated the composition of gut microbiota in AIH patients [96,97,98,99]. Similar to what was previously reported for PBC patients, Wei et al. [96] identified and validated a gut microbiome signature of AIH, including four genera (*Veillonella*, *Lactobacillus*, *Oscillospira*, and *Clostridiales*) that could accurately distinguish AIH patients from healthy controls. Liwinski et al. [98] more recently showed that a disease-specific decline of the relative abundance of Bifidobacterium was observed in AIH patients, and this was associated with failure to achieve remission.

To date, no study has investigated the composition of the microbiome of the biliary tract in patients with AIH and its association with the histological and clinical features of this liver disease.

### 4.3. Microbiota and Primary Sclerosing Cholangitis

Primary sclerosing cholangitis (PSC) is another, even less common, chronic inflammatory disease of the liver and bile ducts that is associated with the development of cholangitis, progressive fibrosis, and end-stage disease requiring liver transplantation [100]. Unlike the above-mentioned diseases, no accepted medical therapy for PSC exists at the moment [101], and this is partly because the etiology and pathogenesis of PSC are not well-understood. As PSC can be associated with intestinal inflammatory liver disease (IBD), especially ulcerative colitis (UC), it has been long hypothesized that the interplay between the gut and hepatobiliary systems played a central role in PSC pathogenesis [102,103], mainly through gut dysbiosis, alteration of intestinal permeability, bacterial translocation, and immune-mediated hepatobiliary inflammation [104,105,106].

In this view, several studies have investigated the gut microbiota composition in PSC patients [107,108,109,110,111,112,113,114,115,116,117], as reported in Table 3. The diversity of PSC is significantly reduced in PSC patients, and its global composition is distinct from that of patients with UC or healthy controls [110,111]; on the other hand, whether there is a difference between PSC patients with and without IBD is controversial [110,113]. At a genus level, the microbiota of PSC patients was overrepresented in *Enterococcus* [111,113,115,116], *Lactobacillus* [111,115,116], *Veillonella* [110,113,114,116], *Streptococcus* [113,115,116], *Fusobacterium* [111], *Rothia* [113], and *Parabacteroides* [116]. In the largest cohort to date, including patients from a German and Norwegian cohort, Kummen et al. [109] not only identified species associated with PSC patients but also a decrease in the abundance of genes related to the synthesis of vitamin B6 and branched-chain amino acids, and this reduction was associated with lower liver-transplantation-free survival. Finally, Lemoinne et al. [114] showed that PSC patients also displayed a fungal gut dysbiosis, with an increased proportion of *Exophiala* and a decreased proportion of *Saccharomyces*.

As for the microbiota of the biliary tract, only a few studies have evaluated its composition by cultures or PCR amplification techniques [91,118,119,120,121,122]. Olsson et al. [118,119] first analyzed the microbiota composition of bile samples from explanted livers of 36 patients with PSC. The authors reported positive cultures in 20 (56%) patients, where *Streptococci* (16/20), *Enterococci* (5/20), and *Staphylococci* (5/20) were the most common bacteria; however, these results might have been biased by culture contamination, recent ERCP, and the use of antibiotics [119]. In the above-mentioned study by Hiramatsu et al. [91], bacterial cultures were positive in one out of five patients (*Streptococcus milleri*), and PCR amplification identified bacteria species in two patients (*S. milleri* and *S. aureus*). In the more recent study by Pereira et al. [121], including bile samples from 80 PSC patients and 46 controls undergoing ERCP, the most common genera in the bile tract were *Prevotella*, *Streptococcus*, *Veillonella*, *Fusobacterium*, and *Haemophilus*.

From a clinical point of view, given this substantial evidence linking gut microbiota and autoimmune cholangitis, different trials have investigated the therapeutic role of different antibiotics [101], such as metronidazole [123,124], vancomycin [123,124,125], and rifaximin [126], with inconclusive results. Of note, in their pilot study, Allegretti et al. [127] showed that fecal microbiota transplantation (FMT) was safe in 10 patients with PSC and concomitant IBD; a decrease in the alkaline phosphatase levels ≥50% was seen in 3 (30%) patients, and this correlated with the bacterial diversity and engraftment after FTM.

In conclusion, PSC is a rare disease with a complex pathogenesis, where gut and biliary tract dysbiosis may play a significant role. The therapeutic role of microbiome modulation in these patients is an intriguing field of research that needs further investigation.

## 5. Microbiome and Liver Transplantation

Liver transplantation (LT) is associated with an overall medical and functional recovery for most patients with end-stage liver disease, and changes in gut microbial composition are no exception. We reported the main evidence on this topic in Table 4. Several studies [128,129,130,131] have shown that at 6 months after LT, the gut microbiota is significantly improved, with increased diversity, increased beneficial autochthonous taxa, such as Ruminococcaceae, Lachnospiraceae, and *Akkermansia*, and a reduction in the pathogenic genera belonging to Enterobacteriaceae (i.e., *Escherichia*, *Shigella*, and *Salmonella*). However, up to 30% of the patients showed persistent cognitive impairment, and this was associated with a relative abundance in Proteobacteria after LT.

Recent developments have shown that the microbiota plays a significant role in the development of complications after LT, where a complex interaction between immunosuppression, antibiotic therapy, infections, and liver allograft immunity takes place [132]. Animal studies have shown that gut bacteria shed microbial-associated molecule patterns into the portal venous circulation, shaping the number, functional activity, and maturational status of liver Kupffer cells [133]. These findings suggest that the gut microbiome is an important modulator of both innate and adaptive liver allograft immunity, and, therefore, its modulation could play a role in the prevention of complications, such as ischemia-reperfusion injury (IRI), acute cellular rejection (ACR), and infections after LT. For instance, both animal [134,135] and human studies [136,137] have shown that probiotic supplementation might attenuate the IRI entity. In fact, cirrhotic patients on rifaximin before LT [136] or patients receiving multi-strain probiotics before LT, in a recent randomized controlled trial (RCT) [137], had a lower incidence of early allograft dysfunction (EAD); this is probably due to the modulation of the intestinal microbiota, the suppression of inflammatory cell activation in the graft, and therefore, the attenuation of the hepatic IRI, the main driver of post-LT organ dysfunction. Regarding ACR, studies in animals [138,139] and humans [140] have established a role of gut dysbiosis in its development, with microbiome alterations found as early as one week after LT. ACR patients presented with a lower microbiome diversity; an increase in Bacteroides, Enterobacteriaceae, *Streptococcaceae*, and *Bifidobacteriaceae*; and a decrease in *Enterococcaceae*, *Lactobacillaceae*, *Clostridiaceae*, and *Ruminococcaceae*. However, a recent meta-analysis failed to show a significant reduction in ACR incidence after probiotic supplementation in LT recipients [141].

It is noteworthy that gut microbiota is a strong predictor of infection after LT, a major cause of morbidity and mortality in these patients [140]. In a recent study [131], a low pre-transplant alpha-diversity, as well as a post-LT abundance of *Enterococcus* and *Klebsiella*, and reductions in *Bacteroides*, *Faecalibacterium*, and *Lachnospira*, were associated with colonization by multidrug-resistance bacteria, a new hallmark of gut dysbiosis. Therefore, the restoration of the intestinal microbiota–host homeostasis has been an attractive strategy to prevent infection after LT, with very promising results. A meta-analysis of four RCTs [141] found that the synergic use of prebiotics and probiotics significantly reduced the infection rates (7% vs. 35%) after LT; these results were later confirmed by a fifth RCT [137]. Moreover, gut dysbiosis was also associated with the development of non-anastomotic biliary strictures [142], hepatic generation after partial liver grafts [132], and the recurrence of liver disease [143].

Finally, only a handful of studies has evaluated the composition of the bile microbiome in LT recipients, exclusively in patients undergoing ERCP/percutaneous transhepatic cholangiography for biliary complications [144]. In a prospective cohort of 213 patients, the authors showed that the most common bacterial isolates were *Enterococcus* spp. (40%), *Streptococci* (20.5%), *Staphylococcus* spp. (12.8%), *Escherichia coli* (10%), and *Klebsiella* spp. (4%); moreover, *Candida albicans* was also found in 15.6% of patients. Importantly, colonization by enteric bacteria, despite successful endoscopic treatment, and fungibilia were associated with lower retransplantation-free survival. Later, the composition of the bile microbiome by was confirmed culture-based techniques by a smaller prospective study [144]. Only in the studies by Liu et al. [145,146] was the bile microbiota assessed by 16S-rRNA gene sequencing. The authors found that the most common genera were *Enterococcus*, *Rhizobium*, *Nevskia*, *Lactococcus*, *Bacillus*, *Clostridium sensu strictu*, *Stenotrophomonas*, *Pseudomonas*, *Streptococcus*, and *Aeromona*. Moreover, the bile microbiome composition differed significantly between patients with clinical symptoms or signs of biliary obstruction and controls, with an increase in Proteobacteria and a decrease in Firmicutes phyla in the case group. In conclusion, these data should be interpreted with caution, as contamination of bile sampling, the use of antibiotics, previous ERCP, etc., are all confounding factors that should be considered. However, the study of the bile microbiome is very promising for a better understanding of the pathogenesis of biliary and non-biliary complications after LT.

## 6. Conclusions and Future Perspectives

The biliary tract is an interesting system interposed between an aseptic system, the hepatocyte, and a rich set of microbes, our gut. The understanding of the interaction of commensal biliary microbes with the host in determining the state of health or disease of the hepatobiliary system is an interesting topic of research, with the aim to define the concept of “biliary dysbiosis”.

While there is some evidence for an association between specific bacterial signatures and conditions, such as biliary stones, inflammatory disorders, or malignancies, causality is far from proven. In this view, therapies modulating the microbiota, with antibiotics, prebiotics, probiotics, and even FMT are intriguing possibilities to be tested in large prospective randomized studies for the management of these diseases.

## Figures and Tables

**Table 1 microorganisms-10-00312-t001:** Gut and biliary microbiota in gallstone disease.

	Potential Pathogenesis Mechanism	Phylum	Family	Genus
Gut Microbiota In Gallstone Disease	Genetic/environmental factors, drugs, lifestyle, comorbidities Cholesterol/bilirubin hypersecretion Intestinal, metabolic or dietary factors Bacterial damaging effect Perturbation of gut microbiota: - Alteration of bile acids metabolism and host metabolic pathways - Release of proinflammatory and vasoactive substances - Role in energy intake, intestinal permeability, promotion of chronic pro-inflammatory states	↑ Firmicutes	Lactobacillaceae Clostridiaceae Ruminococcaceae Acidaminococcaceae Lachnospiraceae	*Clostridium* *Dorea* *Ruminococcus* *Oscillospira* *Veillonella* *Blautia* *Anaerostipes*
		↑ Actinobacteria	Bifidobacteriaceae	*Bifidobacterium*
		↑ Bacteroidetes	Bacteroideceae	*Prevotella* *Bacteroides*
		↑ Fusobacteria	Fusobacteriaceae	*Fusobacterium*
		↑ Proteobacteria		
		↓ Firmicutes	Clostridiaceae Eubacteriaceae Lachnospiraceae	*Faecalibacterium* *Eubacterium* *Lachnospira* *Roseburia*
		↓ Proteobacteria	Desulfovibrionaceae	*Desulfovibrio*
		↓ Actinobacteria	Bifidobacteriaceae	*Bifidobacterium*
		↓ Bacteroidetes	Bacteroideceae Rikenellaceae Paludibacteraceae Barnesiellaceae Muribaculaceae	*Prevotella* *Bacteroides* *Alistipes* *Paludibacter* *Barnesiella*
Biliary Microbiota In Gallstone Disease	Genetic/environmental factors, drugs, lifestyle, comorbidities Cholesterol/bilirubin hypersecretion Intestinal, metabolic or dietary factors Bacterial damaging effect Perturbation of gut microbiota: - Alteration of bile acids metabolism and host metabolic pathways - Release of proinflammatory and vasoactive substances - Role in energy intake, intestinal permeability, promotion of chronic pro-inflammatory states	↑ Proteobacteria	Enterobacteriaceae	
		↑ Firmicutes	Enterococcaceae	*Enterococcus*
		↓ Bacteroidetes		
		↓ Synergistetes	Synergistaceae	*Pyramidobacter*

**Table 2 microorganisms-10-00312-t002:** Summarizing the main microorganisms involved in cholangiocarcinoma and gallbladder cancer.

	Potential Pathogenesis Mechanism	Phylum	Family	Genus
Gut Microbiota in Cholangiocarcinoma	Direct impact of bacterial toxins/metabolites on cancer initiation and growth Modulation of the host local and systemic immune response Alteration of microbial and host metabolism Interaction of gut microbiota on the bile acids metabolism pathways	↑ Firmicutes	Lactobacillaceae Peptostreptococcaceae Ruminococcaceae	*Lactobacillus*
		↑ Actinobacteria	Bifidobacteriaceae	*Actinomyces Alloscardovia*
Biliary Microbiota in Cholangiocarcinoma	Direct impact of bacterial toxins/metabolites on cancer initiation and growth Modulation of the host local and systemic immune response Alteration of microbial and host metabolism Interaction of gut microbiota on the bile acids metabolism pathways	↑ Proteobacteria	Enterobacteriaceae Methylophilaceae Sinobacteriaceae Helicobacteracae Erythrobacteraceae	*Klebsiella* *Helicobacter* *Novosphingobium*
		↑Gemmatimonadetes		
		↑ Nitrospirae		
		↑ Chloroflexi		
		↑ Latescibacteria		
		↑ Planctomycetes		
		↑ Actinobacteria	Bifidobacteriaceae	*Actinomyces*
		↑ Firmicutes	Enterococcaceae Streptococcaceae Bacillaceae	*Dialister* *Streptococcus* *Geobacillus* *Anoxybacillus*
		↑ Bacteroidetes	Bacteroideceae	*Prevotella* *Bacteroides*
		↑ Fusobacteria	Fusobacteriaceae	*Fusobacterium*
		↑ Synergistetes	Synergistaceae	*Pyramidobacter*
		↑ Deinococcus-Thermus	Thermaceae	*Meiothermus*
Biliary Microbiota in Gallbladder Cancer	Direct impact of bacterial toxins/metabolites on cancer initiation and growth Modulation of the host local and systemic immune response Alteration of microbial and host metabolism Inflammation-induced carcinogenesis	↑ Proteobacteria	Enterobacteriaceae	*Salmonella*
		↑ Firmicutes	Peptostreptococcaceae Enterococcaceae	*Peptostreptococcus* *Enterococcus*
		↑ Bacteroidetes		
		↑ Fusobacteria	Fusobacteriaceae	*Fusobacterium*

**Table 3 microorganisms-10-00312-t003:** Main gut and biliary microbiota components involved in autoimmune liver disease.

	Potential Pathogenesis Mechanism	Phylum	Family	Genus
Gut Microbiota in PBC	Interaction between immune and biliary pathways Genetic and environmental factors Molecular mimicry between host antigens and microbe	↑ Proteobacteria		
		↑ Fusobacteria		
		↓ Bacteroidetes		
Biliary Microbiota in PBC	Interaction between immune and biliary pathways Genetic and environmental factors Molecular mimicry between host antigens and microbe	↑ Firmicutes	Staphylococcaceae Enterococcaceae Streptococcaceae Lactobacillaceae	*Staphylococcus* *Enterococcus* *Streptococcus* *Lactobacillus*
Gut Microbiota in PSC	Gut dysbiosis Alteration of intestinal permeability → bacterial translocation Immune-mediated hepatobiliary inflammation	↑ Firmicutes	Enterococcaceae Streptococcaceae Lactobacillaceae Acidaminococcaceae	*Enterococcus* *Streptococcus* *Lactobacillus* *Veillonella*
		↑ Fusobacteria	Fusobacteriaceae	*Fusobacterium*
		↑ Actinobacteria	Micrococcaceae	*Rothia*
		↑ Bacteroidetes	Tannerellaceae	*Parabacteroides*
Biliary Microbiota in PSC	Gut dysbiosis Alteration of intestinal permeability → bacterial translocation Immune-mediated hepatobiliary inflammation	↑ Firmicutes	Staphylococcaceae Enterococcaceae Streptococcaceae Acidaminococcaceae	*Staphylococcus* *Enterococcus* *Streptococcus* *Veillonella*
		↑ Bacteroidetes	Bacteroideceae	*Prevotella*
		↑ Fusobacteria	Fusobacteriaceae	*Fusobacterium*
		↑ Proteobacteria	Pasteurellaceae	*Haemophilus*
Gut Microbiota in AIH	Increased intestinal permeability and gut microbiome dysbiosis	↑ Firmicutes	Acidaminococcaceae Lactobacillaceae Ruminococcaceae	*Veillonella* *Lactobacillus* *Oscillospira*

**Table 4 microorganisms-10-00312-t004:** LT-related gut and bile microbiota alternations.

	Potential Pathogenesis Mechanism	Phylum	Family	Genus
Gut Microbiota in Liver Transplant	Interaction between immunosuppression, antibiotic therapy, infections, and liver allograft immunity Modulation of both innate and adaptive liver allograft immunity → prevention of complications such as ischemia-reperfusion injury, acute cellular rejection, and infections after LT.	↑ Firmicutes	Ruminococcaceae Lachnospiraceae Streptococcaceae Enterococcaceae Lactobacillaceae Clostridiaceae	*Enterococcus*
		↑ Verrucomicrobia	Verrucomicrobiaceae	*Akkermansia*
		↑ Proteobacteria	Enterobacteriaceae	*Klebsiella*
		↑ Bacteroidetes	Bacteroidaceae	*Bacteroides*
		↑ Actinobacteria	Bifidobacteriaceae	
		↓ Firmicutes	Enterococcaceae Clostridiaceae Lachnospiraceae	*Faecalibacterium*
		↓ Proteobacteria	Enterobacteriaceae	*Escherichia* *Shigella* *Salmonella*
		↓ Bacteroidetes	Bacteroidaceae	*Bacteroides*
Biliary Microbiota in Liver Transplant	Interaction between immunosuppression, antibiotic therapy, infections, and liver allograft immunity Modulation of both innate and adaptive liver allograft immunity → prevention of complications such as ischemia-reperfusion injury, acute cellular rejection, and infections after LT.	↑ Firmicutes	Enterococcaceae Streptococcaceae Staphylococcaceae Clostridiaceae	*Enterococcus* *Streptococcus* *Lactococcus* *Staphylococcus* *Clostridium*
		↑ Proteobacteria	Enterobacteriaceae Rhizobiaceae Xanthomonadaceae Pseudomonaceae Aeromonadaceae	*Escherichia* *Klebsiella* *Rhizobium* *Nevskia* *Stenotrophomas* *Pseudomonas* *Aeromonas*
		↑ Ascomycota	Saccharomycetaceae	*Candida*
		↓ Proteobacteria	Enterobacteriaceae	*Escherichia* *Shigella* *Salmonella*
		↓ Bacteroidetes	Bacteroidaceae	*Bacteroides*
		↓ Firmicutes	Clostridiaceae Lachnospiraceae	*Faecalibacterium Lachnospira*

## Data Availability

Not applicable.

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
