# Peer review of "Biliary Diseases from the Microbiome Perspective: How Microorganisms Could Change the Approach to Benign and Malignant Diseases"

_microorganisms, 2022, doi:10.3390/microorganisms10020312_

Round 1

Reviewer 1 Report

The submitted manuscript is an interesting and comprehensive review regarding “biliary dysbiosis” in the aspects of biliary stones, cancer, and inflammation disorders. Overall, the manuscript is well prepared. However, there are some minor issues need to be addressed:

Major Concern:

  1. The tables are kind of confusing and misaligned. Maybe it is the PDF issues. In several tables the brackets under "Family” are not pointing to a phylum. Also, in some tables the “species” are left blank. I would suggest taking that column off.
  2. Authors are recommended to add a column as “Potential Pathogenesis Mechanism” into tables to point out possible mechanisms. Authors can leave “unknown” in some of the part if necessary. Otherwise, simple names of the microbiota are not interesting to readers.

Minor issues:

Line 242: The sentence is incomplete.

Line 441: please correct the word ‘firs’.

Author Response

We thank the reviewer for his revision and comments. We reviewed the tables correcting the misalignment; we took the “species” column off and, as suggested, we added a column as “Potential Pathogenesis Mechanism”. We also revised the minor issues indicated.

Reviewer 2 Report

Binda et al have created an incredibly thorough, well written review on microbes in biliary disease.  Very few reviews focus on this topic and this review fills a gap in overall knowledge on the subject. I applaud the authors for highlighting the difference between microbiota and microbiome and introducing newer terms such as the oncobiome. The tables are excellent, and the authors have extensively reviewed the literature. Overall this review is comprehensive and logical. As a result, I have only minor comments.

Comments:

  1. The authors have correctly not italicized the phylum, classes, families or genera- but there are few instances where the species are included and they need to be italicized. For example- Escherichia coli in line 118, Pseudomonas aeruginosa and Enterococcus faecalis in line 138, etc. Please go through the manuscript and italicize any microbes where species are included.
  2. According to the International Code of Nomenclature of Bacteria, when the genus is written alone, it should not be italicized unless a species name or spp. follows. However, many microbiologists italicize bacterial genera even when written alone (ie. Bacteroides, Clostridium,). Currently, the manuscript does not have the genera italicized in the text, but the genera are italicized in the tables. I don’t have a strong preference and think it’s acceptable either way- but choose one option (genera italicized or not italicized) and be consistent in the manuscript.
  3. Table 1 in line 152 is written as an I instead of 1.
  4. In table 1, there needs to be a space between Bifidobacterium and Prevotella

Author Response

We thank the reviewer for the suggestions. We went through the whole manuscript and italicized both genera and species. We also revised the little mistakes indicated.
